

# Validation of internal reference genes for relative quantitation studies of gene expression in human laryngeal cancer

Xiaofeng Wang[1], Jinting He[2], Wei Wang[1], Ming Ren[3], Sujie Gao[4], Guanjie Zhao[5], Jincheng Wang[3] and Qiwei Yang[3,6]

[1] Department of Stomatology, China–Japan Union Hospital, Jilin University, Changchun, China
[2] Department of Neurology, China–Japan Union Hospital, Jilin University, Changchun, China
[3] Department of Orthopedics, Second Hospital, Jilin University, Changchun, China
[4] Department of Anesthesiology, China–Japan Union Hospital, Jilin University, Changchun, China
[5] Department of Nephrology, China–Japan Union Hospital, Jilin University, Changchun, China
[6] Central Laboratory of Second Hospital, Jilin University, Changchun, China

## ABSTRACT

**Background.** The aim of this study was to determine the expression stabilities of 12 common internal reference genes for the relative quantitation analysis of target gene expression performed by reverse transcription real-time quantitative polymerase chain reaction (RT-qPCR) in human laryngeal cancer.

**Methods.** Hep-2 cells and 14 laryngeal cancer tissue samples were investigated. The expression characteristics of 12 internal reference gene candidates (18S rRNA, GAPDH, ACTB, HPRT1, RPL29, HMBS, PPIA, ALAS1, TBP, PUM1, GUSB, and B2M) were assessed by RT-qPCR. The data were analyzed by three commonly used software programs: geNorm, NormFinder, and BestKeeper.

**Results.** The use of the combination of four internal reference genes was more appropriate than the use of a single internal reference gene. The optimal combination was PPIA + GUSB + RPL29 + HPRT1 for both the cell line and tissues; while the most appropriate combination was GUSB + RPL29 + HPRT1 + HMBS for the tissues.

**Conclusions.** Our recommended internal reference genes may improve the accuracy of relative quantitation analysis of target gene expression performed by the RT-qPCR method in further gene expression research on laryngeal tumors.

## INTRODUCTION

Real-time quantitative polymerase chain reaction (RT-qPCR) is an accurate, sensitive, and rapid method for gene expression studies (*Huggett et al., 2005*). It is considered as the gold standard for gene expression studies. Among all analysis strategies, relative quantification is a relatively simple and common method that is widely used for investigating gene expression in biomedical studies (*Dheda et al., 2004*). An internal reference gene is considered as a stably expressed gene in all biological samples and is used as an internal control to determine the expression levels of target genes. Therefore, the accurate determination of

Corresponding authors
Jincheng Wang,
jinchengwang@hotmail.com
Qiwei Yang, qiweiy@163.com

gene expression levels depends on the selection of a reliable internal reference gene for normalization (*Liu et al., 2014*). The identification of appropriate internal reference genes is a crucial step for relative quantification analysis. An ideal internal reference gene should be universally stable under various experimental conditions (*Derveaux, Vandesompele & Hellemans, 2010*; *Radonic et al., 2004*). In general, internal reference genes, such as 18S rRNA, GAPDH, and ACTB, are chosen for relative quantification analysis between clinical samples. However, increasing evidence has suggested that the expression levels of these commonly used internal reference genes are variable in distinct tissues, cell lines, between treatments of the same cell line (*Ali et al., 2015*; *Ma et al., 2015*; *Yang et al., 2014*; *Yu et al., 2015*), as well as across cell types (*He et al., 2015*; *Li et al., 2015a*; *Li et al., 2015b*). Thus, with the advancement of precise medicine, it is of high importance to evaluate and validate internal reference genes for the target gene expression profile studies among different cell types and tissues.

Laryngeal cancer is a common squamous cell carcinoma that can develop in any part of the larynx. The cure rate of laryngeal cancer is affected by the tumor location. Accordingly, for the purpose of tumor staging, the larynx is divided into three anatomical regions: the glottis, the supraglottis, and the subglottis (*Lozano et al., 2012*). With the great advancement of functional genomics and proteomics studies in cancer research, personalized medicine has become possible. However, personalized treatment of a disease, especially cancer, relies on the identification and validation of the drivers for the disease. However, there is currently no previous research on the selection of suitable internal reference genes for relative quantification analysis of biomarker expression in human laryngeal cancer cell lines and tissues.

A number of genes including 18S rRNA, GAPDH, ACTB, HPRT1, RPL29, HMBS, PPIA, ALAS1, TBP, PUM1, GUSB, and B2M are considered as optimal internal reference genes for relative quantification analysis in other cancers (*Huan et al., 2012*; *Ohl et al., 2005*). In order to determine the optimal internal reference genes for relative quantification analysis of human laryngeal cancer by RT-qPCR, we validated these 12 candidate genes for gene expression studies in a human laryngeal cancer cell line and tissues.

## MATERIALS AND METHODS

### Human laryngeal cancer cell line

Human laryngeal cancer Hep-2 cells were provided by Jilin Cancer Hospital (Changchun, China). The cells were cultivated in Iscove's modified Dulbecco's media containing 10% fetal bovine serum and 100 units/mL penicillin at 37 °C with 5% $CO_2$.

### Laryngeal cancer tissue samples

A total of 14 laryngeal cancer tissue samples were provided by the Tissue Bank of China–Japan Union Hospital, Jilin University (Changchun, China). The Ethics Committee of the China–Japan Union Hospital has a detailed understanding of and approved this study. Written consent was obtained from each patient.
**Table 1** Clinicopathological characteristic of patients.

| Clinicopathological characteristic | Patients with laryngeal cancer |
| --- | --- |
| Age (mean ± standard deviation) | 56.64 ± 4.86 |
| Gender | |
|     Male | 12 |
|     Female | 2 |
| Histopathological type | |
|     Squamous cell carcinomas | 14 |
|     Adenocarcinoma | 0 |
| TNM stage[a] | |
|     Stage 0 | 1 |
|     Stage I | 4 |
|     Stage II | 4 |
|     Stage III | 4 |
|     Stage IV | 1 |

**Notes.**
[a] According to the Union for International Cancer Control.

## RNA extraction and cDNA synthesis

Total RNA was extracted from Hep-2 cells and each tissue sample using TRIzol reagent (Invitrogen, Carlsbad, CA, USA), according to the manufacturer's instructions. The residual genomic DNA was eliminated by DNase I. The concentration, purity, and integrity of the isolated RNA were determined by a NanoDrop 2000 instrument (Thermo Scientific, Waltham, MA, USA) by measuring the absorbance values at 260 nm (A260) and 280 nm (A280). One microgram of total RNA was used for the cDNA synthesis reaction using a cDNA Synthesis kit (Sangon, Biotech, Shanghai, China), according to the manufacturer's instructions.

## RT-qPCR

The sequences of the primers of 12 internal reference genes were chosen based on previous studies (*Battula et al., 2007*; *Mane et al., 2008*) and synthesized by Sangon Biotech (Shanghai, China) as listed in Table 1. RT-qPCR analysis using 2× SG Fast qPCR Master Mix (Sangon Biotech, Shanghai, China) was performed on a LightCycler 480 instrument (Roche, Basel, Switzerland), as described previously (*Ma et al., 2015*; *Yang et al., 2014*; *Yu et al., 2015*). The RT-qPCR was repeated three times for each sample. The cycle threshold value (*Cp* value) data were analyzed using the equation of relative quantities (*Q*): $Q = 2^{-\Delta Cp}$ (*Livak & Schmittgen, 2001*).

## PCR efficiency

Several cDNA samples were selected randomly for the PCR efficiency investigation. cDNA samples were serially diluted from 0.001× to 1×, and the efficiency (*E*) was calculated by the following formula: $E = 10^{-1/\text{slope}}$, where slope stands for the slope of the calibration curve.

## Statistical analysis

Samples were divided into cell line + tissue group and tissue group. Three frequently used software programs, including geNorm (*Vandesompele et al., 2002*) (https://genorm. cmgg.be), NormFinder (*Andersen, Jensen & Orntoft, 2004*) (http://moma.dk/normfinder-software), and BestKeeper (*Pfaffl et al., 2004*) (http://www.gene-quantification.de/bestkeeper.html), were utilized to evaluate the stability of the internal reference genes as described previously (*Ali et al., 2015*; *Ma et al., 2015*; *Yang et al., 2014*; *Yu et al., 2015*).

# RESULTS

## Specificity and efficiency of the amplification reactions of the 12 candidate internal reference genes

A260 and A280 were used to assess the concentration, purity, and integrity of the isolated RNA. The concentration of the isolated RNA was 3087.88 ± 2849.46 ng/µL, with an A260/A280 ratio of 1.94 ± 0.04. Agarose gel electrophoresis and melting curves were used to verify the specificity and efficiency of the amplification reactions. The sizes of the amplified products were in agreement with the expected results, and nonspecific bands were not observed (Fig. 1A). All curves of the amplified fragments exhibited a single signal peak, as shown by the melting curve (Fig. 1B). The PCR efficiency range was between 1.92 and 2.07, and all correlation coefficients were greater than 0.97. These data demonstrated that the specificity and efficiency of the amplification reactions were reliable enough for further analysis.

## The expression levels of candidate internal reference genes

The *Cp* value value was employed to assess the expression levels of the candidate internal reference genes. A lower *Cp* value indicates a higher expression level. The *Cp* values represented in this research from all samples ranged between 7.82 and 29.45. In both groups, 18S had the lowest mean *Cp* values of 9.57 ± 0.96 and 9.62 ± 1.03, and HMBS had the highest mean *Cp* values of 27.15 ± 1.93 and 27.68 ± 1.67; however, the absolute maximum *Cp* value was presented by PUM1, which was 29.45 in each group (Fig. 2).

## The expression stability of the candidate internal reference genes

To better evaluate the expression stability of the internal reference genes, geNorm, NormFinder, and BestKeeper software programs were utilized. According to the results of geNorm, in the cell line + tissue group, RPL29 and HPRT1 had the lowest *M*-values, followed by PPIA, suggesting that these are the most stable internal reference genes for the study of human laryngeal cancer cell lines and tissues. In the tissue group, PPIA and RPL29 had the lowest *M*-values, followed by HPRT1, which suggests that these same genes are the most stable internal reference genes for the study of human laryngeal cancer tissues (Fig. 3A). At least five internal reference genes should be combined to achieve satisfactory accuracy for both groups, with *V*5/6 of 0.145 and 0.132 in the cell line and tissue, respectively (Fig. 3B).

According to the results determined by the NormFinder program, the combination of PPIA and GUSB was the optimal combination in the cell line + tissue group, while PPIA

Peer⌡

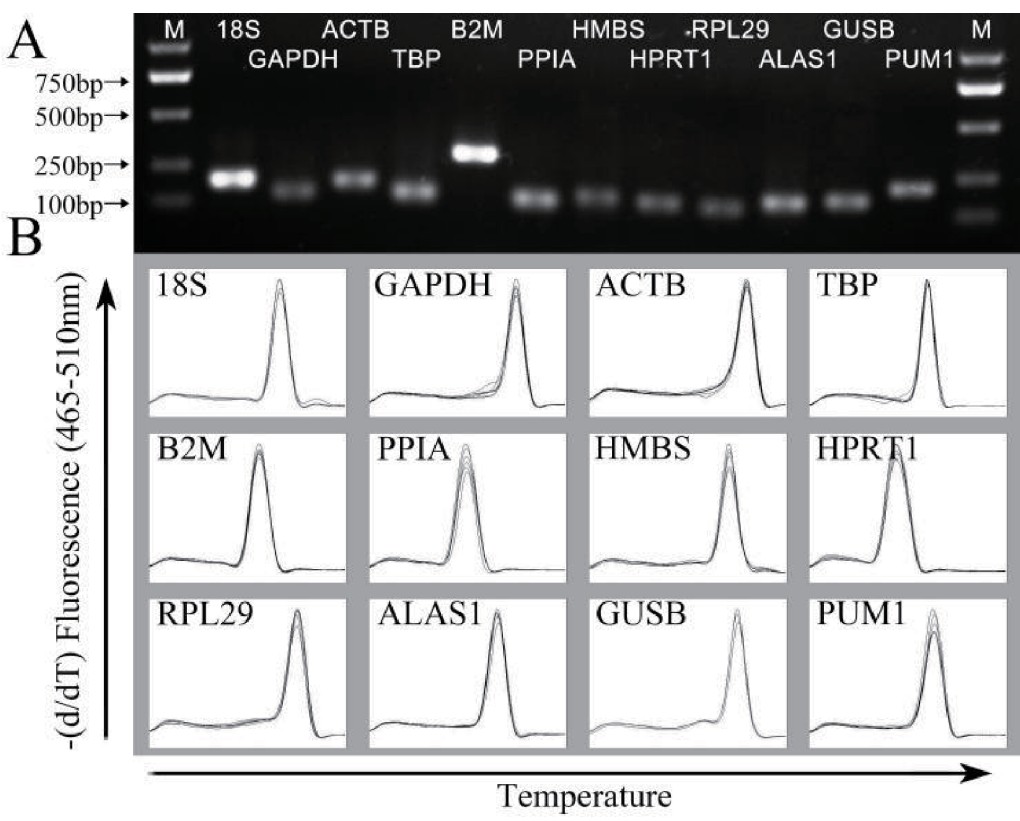

**Figure 1** **Specificity of RT-qPCR amplification.** (A) 1% agarose gel electrophoresis of RT-qPCR amplification products for each of the internal reference genes. The amplified product showed the expected size and no primer dimers. (B) Melting curve analysis for each of the internal reference genes. One single peak was obtained in each amplification reaction. M, marker.

was the optimal internal reference gene in this group, followed by RPL29. In the tissue group, GUSB + RPL29 was the most stable internal reference gene combination, while RPL29 was the most stably expressed gene, followed by HPRT1 (Fig. 4).

The limitation of the BestKeeper program is that only 10 internal reference genes can be assessed at one time. Therefore, we needed to remove the two least stable internal reference genes as indicated by the geNorm program from each group before analysis. In terms of the *R*-value, the optimal internal reference gene in the cell line + tissue group was HMBS, followed by RPL29, GUSB, and HPRT1; while in the tissue group, the optimal internal reference gene was TBP, followed by HMBS, RPL29, and PPIA (Fig. 5).

## DISCUSSION

For the relative quantitative analysis of gene expression, the stability of internal reference genes is crucial for its accuracy. In fact, the expression levels of common internal reference genes vary significantly under various experimental conditions and with different samples (*Ma et al., 2015*; *Yang et al., 2014*; *Yu et al., 2015*). To determine laryngeal cancer gene expression profiles, it is necessary to confirm stable and reliable internal reference genes for RT-qPCR.

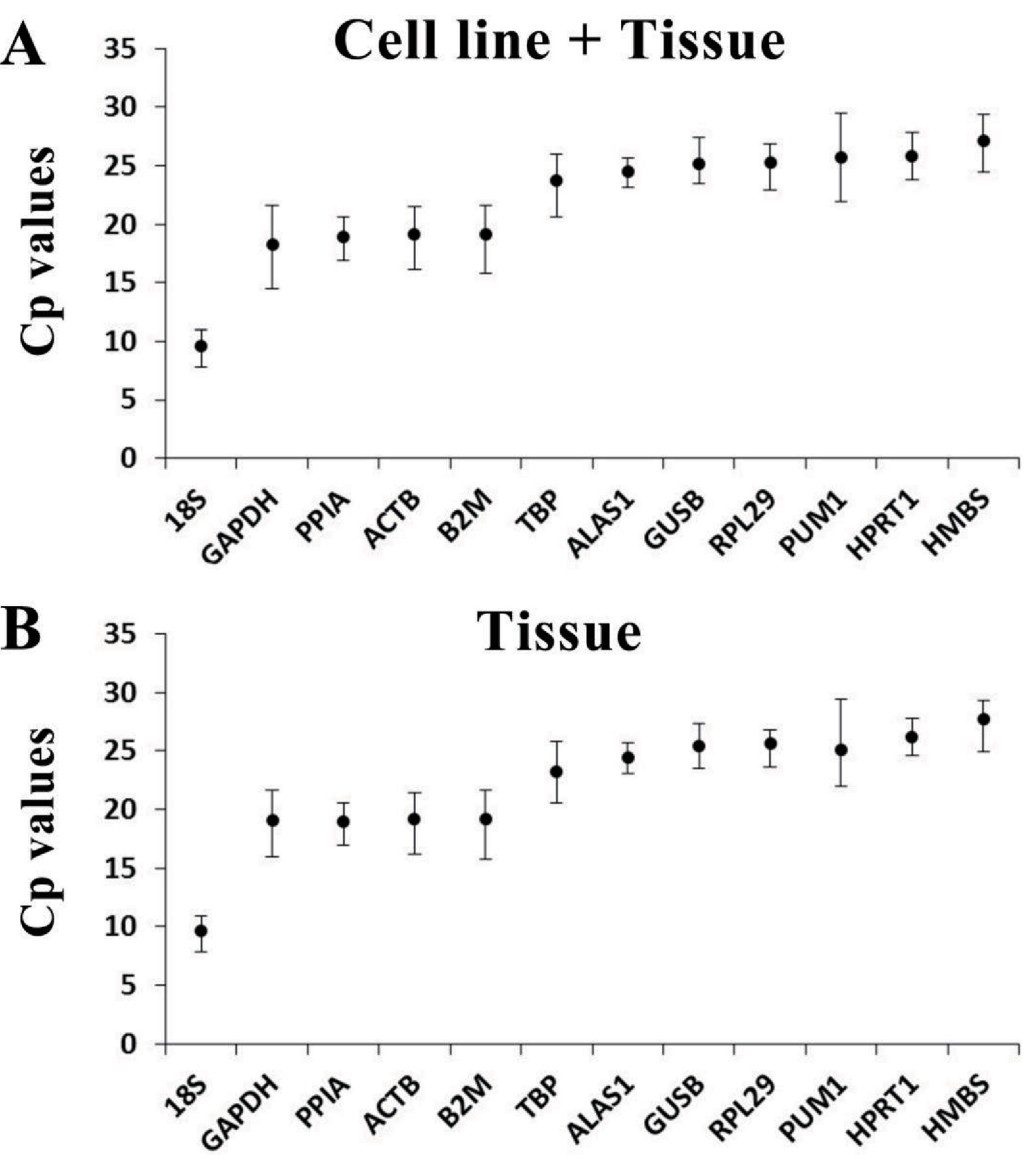

**Figure 2 Cp values of the candidate internal reference genes.** Dots represent the mean Cp value; bars represent the mean ± standard deviation. (A) Cp values of each candidate internal reference gene in the cell line + tissue group (n = 15). (B) Cp values of each candidate internal reference gene in the tissue group (n = 14).

In order to select the suitable internal reference genes for relative quantification analysis of human laryngeal cancer, both a cell line and tissues were investigated in this study. The cell line Hep-2, which is the most commonly used laryngeal cancer cell line for *in vitro* studies, was used. Previous studies have shown that the expression levels of the selected internal reference genes are not directly related to the tumor stage or grade (*Ohl et al., 2005*; *Wan et al., 2010*). In the present study, only malignant squamous cell carcinoma biopsy specimens, which have the highest incidence of pathological types of laryngeal cancer,
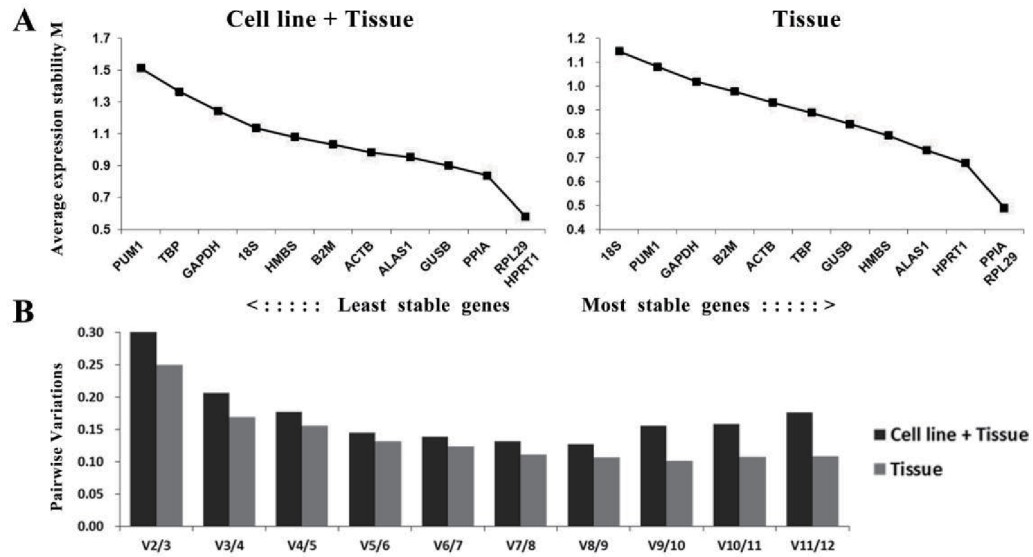

**Figure 3 Analysis results of the geNorm program.** (A) The rank of expression stability of the candidate internal reference genes. The *x*-axis represents the internal reference genes, and the *y*-axis represents *M* values. A lower *M* value represents a higher expression stability ($n = 15$ for the cell line + tissue group; $n = 14$ for the tissue group). (B) Optimal number of internal reference genes in various groups. The *x*-axis represents the number of genes used for the combined analysis, and the *y*-axis represents the pairwise variation value. In $V(n/n + 1)$, *n* is incremented by 1; when the first occurrence of the $V(n/n + 1)$ value is less than 0.15, *n* represents the optimal number of internal reference genes that should be combined to achieve a satisfactory accuracy in the analysis.

were chosen according to the indications for laryngeal cancer surgery. The pathological diagnoses of the biopsy specimens were confirmed by the hospital's Pathology Department.

The expression stability of twelve common internal reference genes, including 18S rRNA, ACTB, GAPDH, HPRT1, RPL29, HMBS, PPIA, ALAS1, TBP, PUM1, GUSB, and B2M, were investigated in the present study. The data were analyzed by the geNorm, NormFinder, and BestKeeper software programs, which were designed to investigate the stability of internal reference genes. According to the geNorm program, RPL29 and HPRT1 were the optimal internal reference genes in the cell line + tissue group. In the tissue group, PPIA and RPL29 were the optimal internal reference genes. Analysis of variance by the NormFinder software program found that PPIA and the combination of PPIA and GUSB in the cell line + tissue group, and RPL29 and the combination of GUSB and RPL29 in the tissue group were the optimal internal reference genes and the best combinations. The BestKeeper program was further used to reduce the one-sidedness of the computational models of the above-mentioned software programs. The results demonstrated that HMBS was the optimal internal reference gene, followed by RPL29 and GUSB in the cell line + tissue group; while TBP, HMBS, and RPL29 were the optimal internal reference genes in the tissue group. Since the rankings of the candidate gene stabilities were slightly different between the two groups, possibly caused by different calculation algorithms (*Bruge et al., 2011*; *Chang et al., 2012*), our data suggest that there is no specific single internal reference gene that can be recommended as the optimal internal reference gene for normalizing

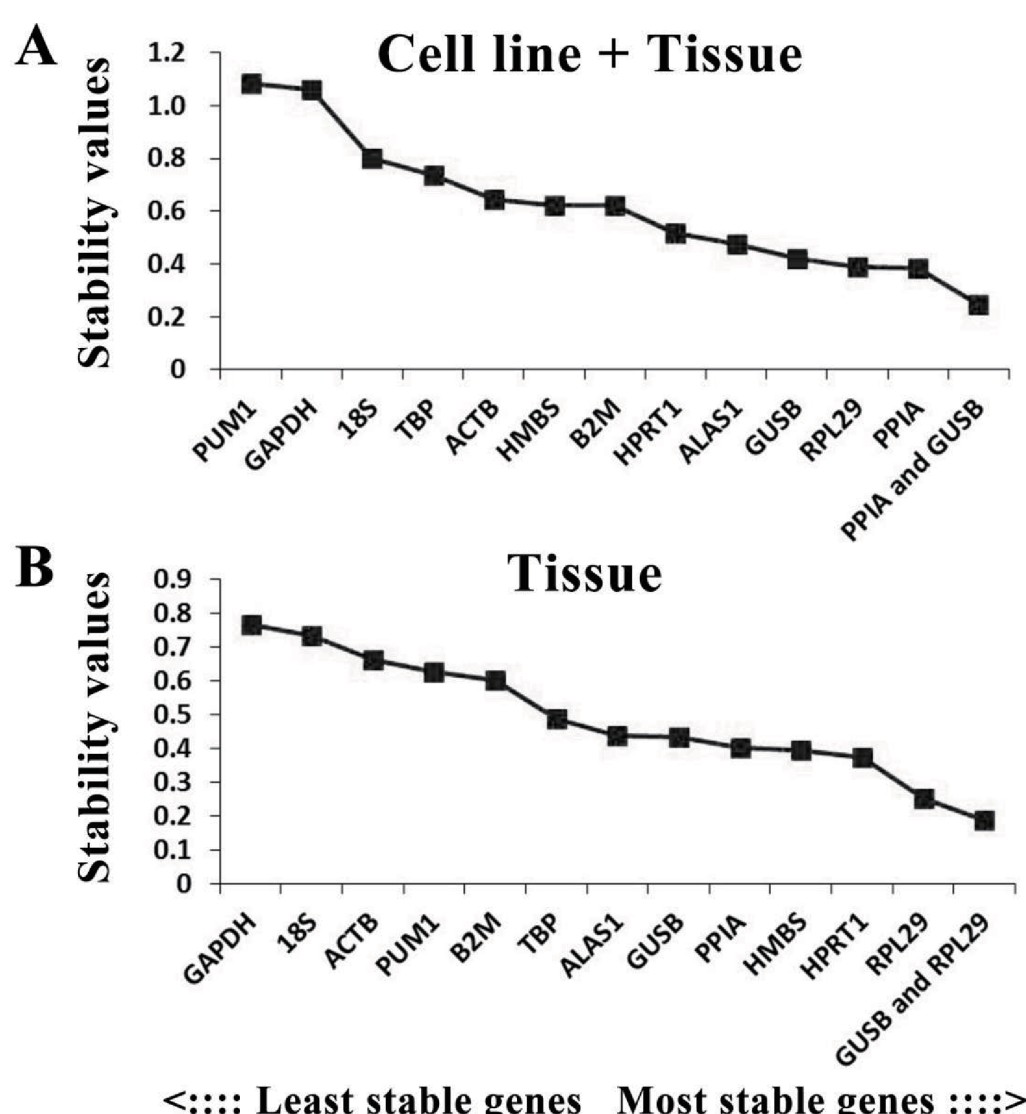

**Figure 4 Analysis results of the NormFinder program.** The *x*-axis represents the internal reference genes, while the *y*-axis represents the stability value. A lower stability value represents higher expression stability. (A) The stability values of each candidate internal reference gene in the cell line + tissue group ($n = 15$). (B) The stability values of each candidate internal reference gene in the tissue group ($n = 14$).

the relative gene quantitation in laryngeal cancer samples. In addition, according to the results of geNorm, the optimal number of internal reference genes used in combination was five in both groups. In contrast, a previous study has suggested that the combination of three internal reference genes is the optimal choice to perform the relative quantitative investigation (*Wisnieski et al., 2013*). Based on the results of the three software programs, we propose that a combination of four internal reference genes is best for normalizing the relative quantitation. Two internal reference genes from the combination suggested by NormFinder and two genes that were top-ranked in all three software programs were

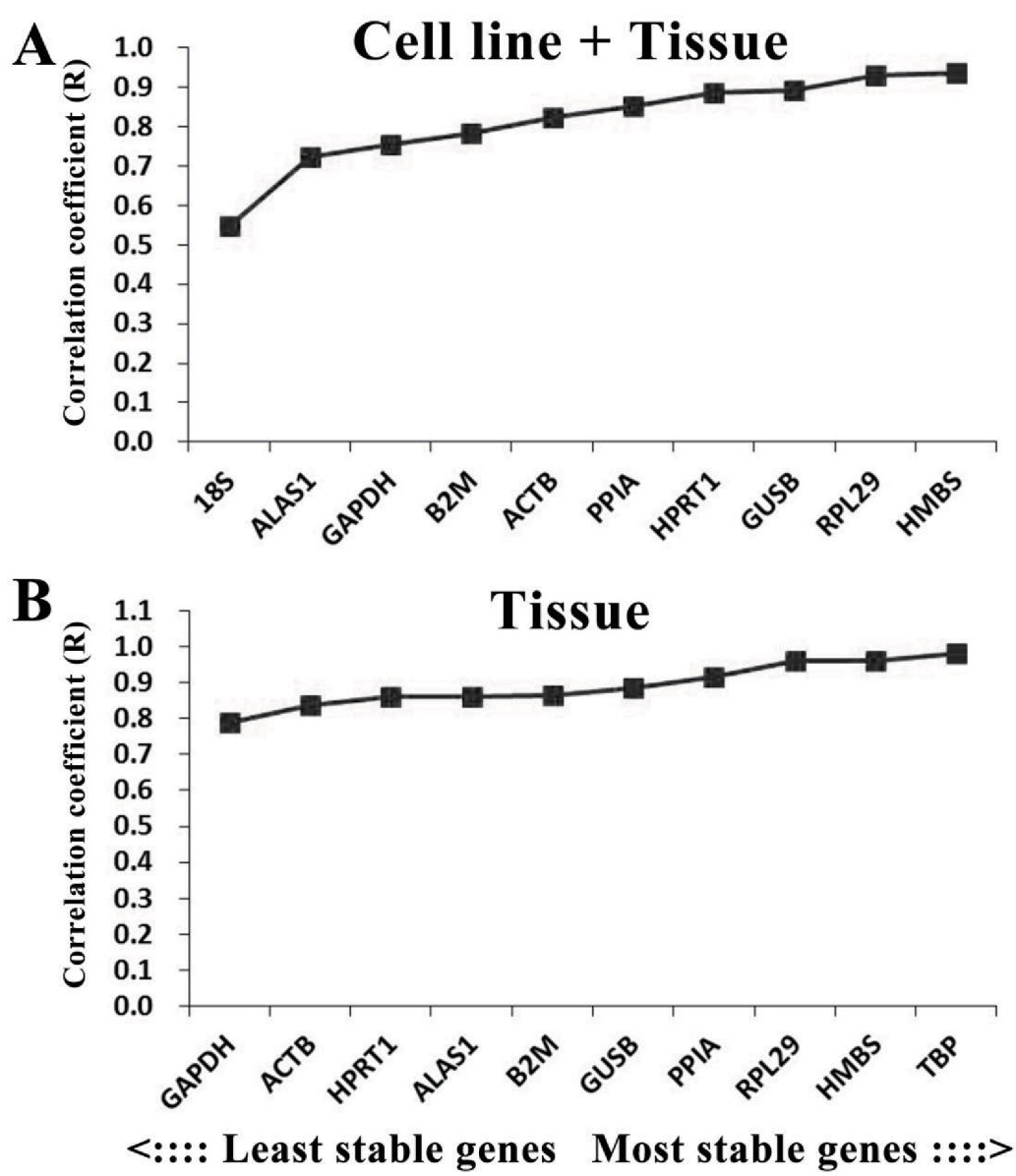

**Figure 5** **Analysis results of the BestKeeper program.** The $x$-axis represents the internal reference genes, while the $y$-axis represents the coefficient of correlation ($R$-value). A higher $R$-value represents higher expression stability. (A) The R-values of each candidate internal reference gene in the cell line + tissue group ($n = 15$). (B) The $R$-values of each candidate internal reference gene in the tissue group ($n = 14$).

selected for the recommended combination. The recommended combination for the cell line + tissue group was PPIA + GUSB + RPL29 + HPRT1; and for the tissue group, it was GUSB + RPL29 + HPRT1 + HMBS.

## CONCLUSION

In summary, the present study evaluated the commonly used internal reference genes for the gene expression profile analysis of human laryngeal cancer cell lines and tissues

from multiple patients and then selected the most appropriate one as well as the optimal combination. Our recommended internal reference genes may improve the accuracy of relative quantitation analysis of target gene expression performed by the RT-qPCR method in further gene expression research on laryngeal tumors. However, in the specific implementation process, it is suggested to further test the internal reference genes recommended according to the specific experimental conditions, as many factors can influence reference gene stability.

### Funding

This study was supported in part by grants from the Natural Science Foundation of China (No. 81503531), the Education Department of Jilin Province (No. 2015536), the Science and Technology Department of Jilin Province (No. 201215078), and the Health and Family Planning Commission of Jilin Province (No. 2011Z048). The funders had no role in study design, data collection and analysis, decision to publish, or preparation of the manuscript.

### Grant Disclosures

The following grant information was disclosed by the authors:
Natural Science Foundation of China: No. 81503531.
Education Department of Jilin Province: No. 2015536.
Science and Technology Department of Jilin Province: No. 201215078.
Health and Family Planning Commission of Jilin Province: No. 2011Z048.

### Competing Interests

The authors declare there are no competing interests.

### Author Contributions

- Xiaofeng Wang conceived and designed the experiments, performed the experiments, analyzed the data, wrote the paper.
- Jinting He performed the experiments, contributed reagents/materials/analysis tools, wrote the paper.
- Wei Wang and Ming Ren contributed reagents/materials/analysis tools.
- Sujie Gao and Guanjie Zhao contributed reagents/materials/analysis tools, prepared figures and/or tables.
- Jincheng Wang analyzed the data.
- Qiwei Yang conceived and designed the experiments, analyzed the data, reviewed drafts of the paper.

### Human Ethics

The following information was supplied relating to ethical approvals (i.e., approving body and any reference numbers):

The Ethics Committee of the China–Japan-Japan Union Hospital has a detailed understanding of and approved this study. Written consent was obtained from each patient.
## Data Availability

The raw data has been supplied as a Supplementary File.

## Supplemental Information

Supplemental information for this article can be found online at http://dx.doi.org/10.7717/peerj.2763#supplemental-information.

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
