# Peer review of "Validation of internal reference genes for relative quantitation studies of gene expression in human laryngeal cancer"

_PeerJ, doi:10.7717/peerj.2763_

## Round 0.1 · original submission · Major Revisions

After reading your manuscript I agree with many of the reviewer comments. As you may not agree with all of the comments, please address each of the points raised by the reviewers in detail with solid justifications.

Reviewer 1 ·

Basic reporting

Figure legends are lacking details. Readers should be able to know what were the tested, on what samples and how many replicates were done by reading the legend.

References are missing in the introduction, in the first half of the first paragraph

Experimental design

RNA purity and integrity analysis (control) have not been detailed (if they were performed). These can greatly influence reference gene expression, and validation (selection) (Huang, X., et al., Placenta, 2013. 34; 544; Lanoix et al., Mol Biotechnol, 2012. 54; 243).

How many replicate where realized?

Validity of the findings

See comments in the previous section (experimental design) concerning RNA purity and integrity.

Comments for the author

While the cell line is from a cancerous tissue, it is perfectly acceptable to use a different combination of reference genes for tissue and for cell lines in the same article. However, it would have been interesting to see if cell treatment or different cell passages affect the evaluation of adequate reference genes in the Hep-2. It is well documented that cell passage and treatment affect housekeeping gene and the internal control genes should be validate with the treatment and for a range on cell passage.

It is noteworthy that the GeNorm V threshold is actually an arbitrary value and should not be taken as absolute. For example 3 genes are at GeNorm V of 0.155 and 5 reference genes are at 0.145, the extra time/cost should be up to the authors to judge if it is relevant to assess 2 extra genes.

As for the conclusion, these results should be taken as a starting point for the reader to test reference genes on laryngial cancer tissue or cell line. While this is a good conclusion, they authors shouls invite readers to use caution and to test the reference genes suggested on their own samples as many factors can influence reference genes stability (sample sex, ethnicity, type of cancer etc.)

·

Basic reporting

Wang and colleagues have performed a detailed analysis of potential internal reference genes for expression assays in laryngeal cancer cell lines and tissue. To be internally consistent and in keeping with current terminology the term “housekeeping gene” should not be used interchangeably with preferred terminology of “internal reference gene.”

Experimental design

This work could benefit from including study of the candidate internal reference genes in benign laryngeal tissue as well as malignant tissue. This would assure that there internal reference gene candidates are not affected by the disease under study.

Validity of the findings

It is not clear as to how the investigators arrived at the two final recommended 4 gene combinations. For example, the NormFinder program is best suited for comparison within and between groups, yet the final recommendation for cell line and tissue were not based on the genes with the best stability values. These final selections need clarification.

Comments for the author

This is a well written manuscript that provides useful information. It can be strengthened by addressing several issues as noted.

---

## Round 0.2 · accepted · Accept

The authors did a good job of addressing the concerns of the reviewers
I have but one small comment. Please replace "quantitation" with "quantification" throughout the manuscript.